**Data Availability Statement:** All relevant data are available on BioStudies through the link: https://

# Validity and reliability of the Thai version of the ASCRS SPEED II preoperative OSD questionnaire in Thai cataract surgery candidates

Kaevalin Lekhanont[1]*, Paphada Ariyanonthacha[1], Chathaya Wongrathanandha[2], Prae Phimpho[1], Nontawat Cheewaruangroj[1]

1 Department of Ophthalmology, Faculty of Medicine Ramathibodi Hospital, Mahidol University, Bangkok, Thailand, 2 Department of Community Medicine, Faculty of Medicine Ramathibodi Hospital, Mahidol University, Bangkok, Thailand

* lekhanont@yahoo.com

## Abstract

### Objectives

To develop a Thai version of the American Society of Cataract and Refractive Surgery (ASCRS)-modified Standard Patient Evaluation of Eye Dryness (SPEED) II© preoperative ocular surface disease (OSD) questionnaire (ASCRS SPEED II-Thai) and evaluate its validity and reliability in Thai cataract surgery candidates.

### Methods

The original English version of the questionnaire was translated and adapted cross-culturally to Thai language. The ASCRS SPEED II-Thai was evaluated for content validity, internal consistency, and test-retest reliability. Five experts in the ocular surface field critically reviewed the ASCRS SPEED II-Thai to measure the content validity indexes (CVI). A total of 105 cataract surgery candidates were recruited from an ophthalmology clinic to complete the questionnaire. Cronbach's alpha and intraclass correlation coefficient (ICC) were calculated to assess internal consistency and test–retest reliability, respectively.

### Results

During the translation and cross-cultural adaptation processes, only two minor modifications were made to the original version for clarification without changing their meaning. All items in the ASCRS SPEED II-Thai had an item-level CVI of 1.00, representing complete agreement among content experts. The scale-level CVI was 1.00, indicating excellent content validity of the questionnaire. The participants found no difficulty in understanding each question in the pilot test. Cronbach's alpha for the ASCRS SPEED II-Thai was 0.82, indicating good internal consistency. The test-retest reliability was good to excellent, with ICC values ranging from 0.83 to 1.00 ($P < 0.001$).

www.ebi.ac.uk/biostudies/studies/S-BSST1369?key=96fb23d2-f605-4615-9cc7-d6d4f5e93196.

**Funding:** The author(s) received no specific funding for this work.

**Competing interests:** The authors have declared that no competing interests exist.

## Conclusions

The ASCRS SPEED II-Thai is a valid clinical tool with adequate reliability for preoperative OSD screening among Thai cataract surgery candidates.

## Introduction

Ocular surface disease (OSD), especially dry eye disease (DED), is one of the most common conditions that lead patients to seek medical eye care [1]. The prevalence and incidence of DED have been found to increase with age and over time [2]. With longer life expectancy and increasing demand for improved quality of life, the number of patients with underlying DED presenting for cataract surgery has been projected to grow continuously. Thus, dry eye symptoms or signs noted in cataract surgery candidates are expected and should not be overlooked [3, 4]. OSD can not only primarily aggravate visual disturbance but also compromise the surgical outcomes of cataract surgery. Impaired ocular surface resulting from OSD adversely affects preoperative measurements, including keratometry, biometry, and topography [5]. Additionally, it may increase surgical difficulty [6]. Meanwhile, cataract surgery itself and postoperative medication regimen can induce or exacerbate OSD, resulting in poorer postoperative outcomes and patient dissatisfaction [5, 7]. Hence, detection of predisposition to DED, early DED, or other non-DED subtypes of OSD prior to cataract surgery is as crucial as other routine preoperative diagnostic assessments, so that any unrecognized OSD is properly managed proactively to optimize visual outcomes after cataract surgery [6].

The American Society of Cataract and Refractive Surgery (ASCRS) Cornea Clinical Committee has developed a novel preoperative OSD algorithm and preoperative OSD questionnaire to identify any form of OSD before surgery, regardless of the presence of suggestive symptoms [5]. Because none of the published validated DED questionnaires were created specifically for preoperative cataract or refractive surgery populations, with permission from Johnson & Johnson Vision, Inc., their validated Standard Patient Evaluation of Eye Dryness (SPEED) questionnaire was amended to include extra questions relevant to identifying OSD in preoperative patients. The new questionnaire is called the ASCRS SPEED II© preoperative OSD questionnaire, which is easily adoptable, non-invasive, time-saving for surgeons, and educational for patients [5].

Asian ethnicity is a mostly consistent risk factor for DED [8]. Previous studies have shown that DED is prevalent in the elderly Thai population [9, 10]. The number of Thai patients with new-onset significant dry eye symptoms after phacoemulsification was also nearly 10% [11]. However, at present, there is no specific or reproducible Thai questionnaire to be used as a preoperative screening tool for patients undergoing cataract surgery. Therefore, the purpose of this study was to develop a Thai version of the ASCRS SPEED II© preoperative OSD questionnaire (ASCRS SPEED II-Thai) and evaluate its reliability and validity for OSD screening in Thai cataract surgery candidates.

## Materials and methods

### Study design

This cross-sectional study was approved by the Research and Ethics Committee of the Faculty of Medicine Ramathibodi Hospital (COA. MURA2021/68), and adhered to the tenets of the Declaration of Helsinki.

## ASCRS SPEED II preoperative OSD questionnaire

The ASCRS SPEED II© preoperative OSD questionnaire contains 20 main questions divided into two parts. The first part, which was derived from the original version of the SPEED questionnaire, comprised two questions related to the frequency and severity of four major dry eye symptoms (dryness, grittiness, or scratchiness; soreness or irritation; burning or watering; and eye fatigue). Patients were asked to rate each symptom on a 4-point scale based on their frequency, from "never" (score 0) to "constant" (score 3). Each symptom severity was further measured using a 5-point scale, where 0 = no problems and 4 = intolerable (unable to perform my daily tasks) [5]. The total SPEED score from 0 to 28 was tallied and recorded at the bottom of the page for easy reference.

The second part consisted of 18 questions, including one question regarding symptom experiences over the past three months; four questions about the use of eye drops for lubrication, the frequency of lubricating eye drops, the presence of fluctuating vision, and its improvement with blinking and/or lubricants; 10 questions about the presence of diagnosis or symptoms of other subtypes of OSD (blepharitis, stye, contact lens-related OSD, allergic conjunctivitis); two questions about whether patients mind wearing eyeglasses and/or contact lenses for improving their vision, and if yes, whether they would be willing to pay out-of-pocket costs to reduce or eliminate their dependence on them; and one last question to describe patients' personality from easygoing to perfectionist. The total number of checked red boxes from 0 to 18 was also tallied and recorded at the bottom of the page. Although the additional questions in the second part do not have a validated scoring rubric, the higher total number of checked red boxes and higher total SPEED score may reflect a greater possibility of visually significant OSD and an increased need for more aggressive preoperative management [5].

## Translation of the ASCRS SPEED II-Thai

With permission from the developers of the ASCRS SPEED II© preoperative OSD questionnaire (Christopher E. Starr, et al, the ASCRS Cornea Clinical Committee, Journal of Cataract and Refractive Surgery, Wolters Kluwer Health, Inc., May 10, 2020), the questionnaire was translated from English to Thai. The translation process followed standard guidelines for cross-cultural adaptation of self-report measures, using a forward-backward procedure, to ensure that the translated questionnaire was equivalent to the original one, meaningful, and applicable in Thai culture [12, 13]. All translators were bilingual in English and Thai and had obtained a minimum qualification of a master's degree.

**Step 1: Forward translation.** Two separate initial translations from English to Thai, referred to as Translation 1 (T1) and Translation 2 (T2), were conducted by two independent translators. T1's translator was a cornea specialist who was aware of the concept that the questionnaire intended to measure so that the translation closely resembled the original one. T2's translator was a professional translator who was unaware of the objective of the questionnaire and was thus more likely to detect subtle differences in meaning in the original questionnaire. Both forward translations were then compared and any discrepancies between T1 and T2 were discussed and resolved between the two translators. A single common forward translation, designated Translation 12 (T12) was produced.

**Step 2: Backward translation.** The T12 version of the questionnaire was independently back-translated into English by two other translators who were completely blinded to the English version, were not aware of the intended concept of the questionnaire measures, and had limited background in ophthalmology. This would avoid bias and potentially reveal misunderstandings, unclear wordings, or unexpected meanings of the items in forward translations. These versions were called back-translation 1 (BT1) and back-translation 2 (BT2).

**Step 3: Expert committee review.** An expert committee was constituted to consolidate all versions of the translations and create a preliminary version of ASCRS SPEED II-Thai for pretesting. Members of the committee included two research methodologists, four translators, and a cornea specialist who had academic and clinical experience with people with DED, so that she was familiar with the construct of interest. The expert committee reviewed all the translations (T1, T2, T12, BT1, and BT2) and determined whether the translated and original versions achieved semantic, idiomatic, experiential, and conceptual equivalence. Any ambiguities, discrepancies, or cultural appropriateness were discussed and resolved to reach a consensus on all items. All issues addressed and the corresponding resolutions were documented in Thai written reports.

## Assessment of psychometric properties of the ASCRS SPEED II-Thai

**Content validity.** Validity is defined as the extent to which an instrument appropriately measures what it is intended to [14]. For this reason, after translation, a content validity test was conducted to ensure that the ASCRS SPEED II-Thai was valid. The recommended number of experts to review an instrument varies from three to ten individuals, but at least five people are preferred to have sufficient control over chance agreement [14, 15]. Hence, five experts on the ocular surface field who had ten or more years of experience were invited to review the preliminary version of the ASCRS SPEED II-Thai. Experts critically reviewed the questionnaire line-by-line by referring to the study objectives and conceptual framework and judged whether all the items were relevant to the target population and the intended purpose of the questionnaire using a content validation form. A 4-point Likert scale was used for the relevance scale, and responses included: 1 = not relevant, 2 = somewhat relevant, 3 = quite relevant, and 4 = highly relevant. Ratings of 1 and 2 were considered invalid, while ratings of 3 and 4 were considered valid [14]. Additional expert comments and suggestions were written on the hard copy of the questionnaire.

**Pilot testing.** The prefinal version of the ASCRS SPEED II-Thai was pilot tested on a small group of 30 cataract surgery candidates in a face-to-face manner. After completing the questionnaire, the investigator asked the participants to elaborate on how well they understood each item in the questionnaire and how well the different response options were reached. This was to confirm that the translated items retained the same meaning as the original items and that there were no confusing questions in the applied situation. Any problematic items were amended as required. The final version of ASCRS SPEED II-Thai was established.

**Reliability test.** Reliability is defined as the extent to which a measurement is consistent and free from error [16]. To evaluate the reliability of the final version of ASCRS SPEED II-Thai, two methods, internal consistency and test-retest reliability, were examined. Internal consistency was defined as the consistency of individual responses across the items on this multiple-item questionnaire. Because all the items in the questionnaire were supposed to measure different aspects of the same underlying OSD, the scores of those different items should have been correlated with each other. Test-retest reliability represented the consistency of individuals' responses to the same questionnaire items administered twice under the same conditions over a period of time.

According to the guidelines for sample size calculation in reliability studies [17, 18], at least 105 cataract surgery candidates had to be recruited for the study. The time lapse between the two questionnaire administrations was 14 days, which was considered long enough to allow the memory effects to fade and to prevent fatigue, but not too long to avoid changes in the measured variables to occur [19].

**Eligibility criteria for participants in the pilot and reliability tests.** Adult Thai patients aged 18 years or older who were scheduled for cataract surgery at the ophthalmology clinic of Ramathibodi Hospital in Bangkok, Thailand, between 21 March 2021 and 20 January 2022

were eligible for participation in the study. Inclusion criteria were literacy in Thai and voluntary willingness to repeat the questionnaire at a later time point. Patients were excluded from the study if they had no light perception vision and the cataract was going to be operated on because it became intumescent or for cosmetic reasons, were diagnosed with retinal or optic nerve diseases that were associated with poor postoperative visual outcomes, required combined cataract surgery, or had any medical conditions that could limit their ability to answer the questions. Written informed consent was obtained from all participants.

## Statistical analysis

Statistical analyses were conducted using STATA version 16 (Stata Corp, College Station, TX, USA). For continuous data, normally distributed variables are presented as means and standard deviations (SD); non-normally distributed variables are presented as medians and ranges. For categorical data, frequencies and percentages were calculated for each variable. Statistical significance was set at $P < 0.05$.

## Content validity

Content validity index (CVI) for each item and the overall questionnaire was used to assess the content validity of the questionnaire. The item-level CVI (I-CVI) was calculated as the number of experts giving a rating of either 'quite relevant' or 'highly relevant' for each item divided by the total number of experts. To indicate that the item is relevant, the I-CVI should be 1.00 when there are five or fewer judges. Subsequently, the CVI for the entire questionnaire or the scale-level CVI (S-CVI) was computed as the number of items in the questionnaire that achieved a rating of 'quite or highly relevant' by the experts. There are two approaches for calculating an S-CVI: universal agreement (UA) among all experts (S-CVI/UA) and the average CVI (S-CVI/Ave). The S-CVI/UA was defined as the proportion of items on the questionnaire that achieved a rating of 'quite or highly relevant' by all the experts, while the S-CVI/Ave was defined as the average proportion of items rated as relevant across the various judges. S-CVI/UA was calculated by adding all items with an I-CVI equal to 1 divided by the total number of items. S-CVI/Ave was calculated by dividing the sum of all I-CVIs by the total number of items. A minimum S-CVI/UA of 0.8 and an S-CVI/Ave of 0.9 or higher indicate excellent content validity [14].

## Reliability

Descriptive statistics were used to describe the baseline demographic characteristics of participants. Internal consistency, which reflected the extent to which all the items in the questionnaire were inter-correlated, was estimated using Cronbach's alpha. A Cronbach's alpha greater than 0.70 was considered adequate internal consistency. Test-retest reliability, which refers to the probability of producing the same results with repeated administration of the questionnaire, was assessed using the intraclass correlation coefficient (ICC). Based on the 95% confidence interval of the ICC estimate, values of at least 0.75 were indicative of good to excellent reliability [16, 19].

## Results

### Translation process

To better match Thai culture and language, the two items were modified for clarification without changing their meaning. In item 3, the additional information "Can select one or more answers" was given in brackets at the end of the question, as in Thai, a multiple-choice question traditionally allows respondents to select only one option from a list of choices. In the

item "Have you had any of these symptoms recently?", the phrase "Crusting around lashes" was changed to "Crusting at lashes", because in Thai, around lashes might refer more to the area of periorbital skin, while at lashes expresses the exact location of eyelashes and eyelid margin. The participants did not report any misunderstandings regarding any of the items.

## Content validity results

All five content experts invited to participate agreed to participate in the study. Three were faculty and guest instructors from the Department of Ophthalmology, Faculty of Medicine Ramathibodi Hospital, Mahidol University. The mean age of the experts was 43.4 (SD = 3.6) years. The I-CVI of each item in the questionnaire was 1.00, demonstrating complete agreement among content experts. Therefore, the calculated S-CVI/UA and S-CVI/Ave were both 1.00, indicating the excellent content validity of the ASCRS SPEED II-Thai.

## Reliability results

**Participant characteristics.** A survey was conducted among 118 cataract surgery candidates to measure the internal consistency and test–retest reliability of the questionnaire. Of the 118 eligible participants, 105 completed questionnaires at both the test and retest assessments and their data were used for all analyses. The initial questionnaires from 13 subjects were rejected because five of them were incompletely filled, and eight subjects were absent on the second day of the survey, resulting in missing data on the retest.

Demographic characteristics of the 105 participants are presented in Table 1. The mean age was 67.7 ± 9.4 years (range, 35–84 years). Most of the participants were female (57/105, 54.3%), retired (82/105, 78.1%), and had chronic underlying medical conditions, with hypertension (62/105, 59.0%), dyslipidemia (54/105, 51.4%), diabetes mellitus (42/105, 40.0%), renal disease (10/105, 9.5%), and cardiovascular disease (9/105, 8.6%) being the most common. The mean BCVA value was 0.58±0.23 logMAR. Six eyes (5.7%) had undergone previous intraocular surgery, including pars plana vitrectomy (5/105) and trabeculectomy (1/105). Frequent concurrent ocular diseases included glaucoma (13.3%), diabetic retinopathy (12.4%), age-related macular degeneration (9.5%) and DED (4.8%). A significant number of participants (47/105, 44.8%) had dry eye symptoms sufficient to use artificial tears regularly without a prior history of DED diagnosis.

According to the validated numerical scoring system of SPEED retained in the ASCRS-modified Preoperative OSD SPEED II© questionnaire [5], 42 (40.0%) had no significant symptoms and were classified as asymptomatic, whereas 25 (23.8%), 18 (17.1%), and 20 (19.0%) were classified as having mild, moderate, and severe DED, respectively.

**Internal consistency and test–retest reliability.** A total of 22 items derived from these 20 questions described DED-related symptoms, severity, and diagnosis or symptoms of other OSD subtypes. However, only 20 items were used to calculate Cronbach's alpha because two items (items 11 and 18) were open-ended. Cronbach's alpha was 0.82 for the whole questionnaire, indicating good internal consistency. The ICC ranged from 0.83 to 0.98, and all values exceeded the recommended level of 0.75, indicating good-to-excellent test-retest reliability (Table 2). Item 21 was automatically dropped from the analysis because it was a constant. Thus, the reliability level of the ASCRS SPEED II-Thai was regarded as good to excellent.

## Discussion

This study aimed to develop and validate the Thai version of the ASCRS SPEED II© preoperative OSD questionnaire for use in initial OSD screening before cataract surgery. Given that failure to diagnose and adequately address OSD prior to cataract and refractive surgery could

**Table 1. Demographic characteristics of 105 patients included in reliability testing.**

| Characteristics | Number (percentage) |
|---|---|
| Age, mean (SD) [range], year | 67.7 (9.4) [35–84] |
| Sex, female (%) | 57 (54.3) |
| Employment status | |
| • Retired | 82 (78.1) |
| • Employed | 18 (17.1) |
| • Unemployed | 5 (4.8) |
| Ocular comorbidities | |
| • None | 54 (51.4) |
| • Glaucoma | 14 (13.3) |
| • Diabetic retinopathy | 13 (12.4) |
| • Age-related macular degeneration (AMD) | 10 (9.5) |
| ○ Dry AMD | 8 (7.6) |
| ○ Wet AMD | 2 (1.9) |
| • Dry eye disease | 5 (4.8) |
| • High myopia | 3 (2.9) |
| • Allergic conjunctivitis | 2 (1.9) |
| • Polypoidal choroidal vasculopathy | 2 (1.9) |
| • Presumed herpes simplex keratitis scar | 1 (0.9) |
| • Fuchs endothelial corneal dystrophy | 1 (0.9) |
| Systemic comorbidities | |
| • None | 15 (14.3) |
| • Hypertension | 62 (59.0) |
| • Dyslipidemia | 54 (51.4) |
| • Diabetes mellitus | 42 (40.0) |
| • Renal disease | 10 (9.5) |
| • Cardiovascular disease | 9 (8.6) |
| • Cancer (breast, colon, cervix) | 5 (4.8) |
| • Gout | 4 (3.8) |
| • Thyroid disease | 4 (3.8) |
| • Rheumatoid arthritis | 3 (2.9) |
| • Knee osteoarthritis | 3 (2.9) |
| • Thalassemia | 3 (2.9) |
| • Benign prostatic hyperplasia | 3 (2.9) |
| • Human immunodeficiency virus infection | 1 (0.9) |
| • Immunoglobulin G4-related disease | 1 (0.9) |
| • Bipolar disorder | 1 (0.9) |

potentially result in either postoperative disappointing vision, new or worsening OSD symptoms, or infection, new point-of-care diagnostic tools such as tear osmolarity and matrix metallopeptidase 9 (MMP-9) tests have been suggested to be integrated into a routine preoperative workflow [5]. Nevertheless, their adoption in presurgical patients in clinical practice has been slow, possibly due to the increased cost and length of preoperative workup. Therefore, the ASCRS Cornea Clinical Committee developed the consensus-based ASCRS preoperative OSD algorithm and SPEED II© preoperative OSD questionnaire to raise awareness of the most current diagnostic tools and aid surgeons in efficiently detecting and managing visually significant OSD before phacorefractive or keratorefractive surgery [5].

In Thailand, since an in-office test of MMP-9 is not commercially available and tear osmolarity test is expensive and non-reimbursable, using the self-report screening questionnaire would be easily adopted and reduce surgeon chair time. Hence, the English version was cross-culturally adapted to ensure that the translated questionnaire was equivalent to the original version, without confusion. This process was conducted according to the standard guidelines for cross-cultural adaptation of self-report measures [12, 13]. Only two changes were made to the original version to make the choices easier for participants to understand.

**Table 2. Ocular surface disease-related item analysis and test-retest reliability of the Thai version of the ASCRS SPEED II preoperative OSD questionnaire.**

| Item | Question | | ICC | 95% CI of ICC | P-value* (F test) | Interpretation** |
|---|---|---|---|---|---|---|
| 1 | Frequency of symptoms | Dryness, grittiness, or scratchiness | 0.910 | 0.868–0.939 | 0.000 | Excellent |
| 2 | | Soreness or irritation | 0.898 | 0.851–0.931 | 0.000 | Good |
| 3 | | Burning or watering | 0.947 | 0.922–0.964 | 0.000 | Excellent |
| 4 | | Eye fatigue | 0.943 | 0.916–0.961 | 0.000 | Excellent |
| 5 | Severity of symptoms | Dryness, grittiness, or scratchiness | 0.925 | 0.890–0.949 | 0.000 | Excellent |
| 6 | | Soreness or irritation | 0.839 | 0.762–0.890 | 0.000 | Good |
| 7 | | Burning or watering | 0.979 | 0.968–0.986 | 0.000 | Excellent |
| 8 | | Eye fatigue | 0.941 | 0.913–0.960 | 0.000 | Excellent |
| 9 | Please check if you have experienced above symptoms | | 0.825 | 0.732–0.885 | 0.000 | Good |
| 10 | Do you use eye drops for lubrication? | | 0.961 | 0.942–0.973 | 0.000 | Excellent |
| 11 | If yes, how often? | | | | | |
| 12 | Do you have fluctuating vision? | | 0.912 | 0.870–0.940 | 0.000 | Excellent |
| 13 | If yes, does the fluctuating vision improve with blinking and/or lubricating drops? | | 0.852 | 0.783–0.900 | 0.000 | Good |
| 14 | Have you been told you have blepharitis? | | 1.000 | N/A | N/A | Excellent |
| 15 | Have you been treated for a stye? | | 0.942 | 0.915–0.961 | 0.000 | Excellent |
| 16 | Have you had any of these symptoms recently? | | 1.000 | N/A | N/A | Excellent |
| 17 | Do you wear contact lenses? | | 0.963 | 0.946–0.975 | 0.000 | Excellent |
| 18 | If yes, when was the last time you wore them? | | | | | |
| 19 | If yes, do your eyes feel worse when they are on? | | 0.957 | 0.937–0.971 | 0.000 | Excellent |
| 20 | Do your eyes itch? | | 1.000 | N/A | N/A | Excellent |
| 21 | If yes, do you have known environmental allergies or allergic conjunctivitis? | | N/A | N/A | N/A | N/A |
| 22 | Are your ocular symptoms symmetric between both eyes? | | 0.948 | 0.924–0.965 | 0.000 | Excellent |

CI = confidence interval, ICC = intraclass correlation coefficient; N/A = not applicable

*A P-value < 0.05 was considered statistically significant.

**ICC values less than 0.50 indicate poor reliability, values between 0.50 and 0.75 indicate moderate reliability, values between 0.75 and 0.90 indicate good reliability, values greater than 0.90 indicate excellent reliability.[16]

Items 11 and 18 are shaded gray because they were open-ended questions so that they were not used to calculate Cronbach's alpha.

Our psychometric results showed that the ASCRS SPEED II-Thai had satisfactory content validity and reliability among Thai cataract surgery candidates. According to the ocular surface experts, this questionnaire demonstrated excellent content validity. This indicates that the ASCRS SPEED II-Thai contains relevant questions that could be applied to identify OSD and probably determine the risk factors associated with the condition. In addition, the verification of reliability with Cronbach's alpha revealed a high total scale of 0.82, denoting good internal consistency. This means that all items were highly correlated and measured the various components of OSD. Furthermore, the ICC, which is an accurate measure of reliability, reflecting both the degree of correlation and agreement between the first and second measurements, was also calculated in this study. All ICC values represented good-to-excellent test-retest reliability of the questionnaire, supporting that the ASCRS SPEED II-Thai was reproducible.

The current study has some limitations. First, it was conducted at a single university hospital, possibly introducing a sample selection bias. Differences in some variables, such as age, socioeconomic status, and education level, were also not accounted for, possibly influencing the participants' responses. Second, only the content validity of the questionnaire, which is the most important aspect of questionnaire validation, was evaluated. Construct validity was not

assessed because there was no preexisting validated Thai questionnaires that measured similar or dissimilar characteristics of OSD, and relationships between symptoms and signs of DED have been known to be low and inconsistent [20]. Considering a combination of the questionnaire and other objective dry eye tests to evaluate other aspects of validity, including construct, criterion, and predictive validity, merits further investigation. Third, for the test-retest study, the typical concern of all self-reported measures was the susceptibility to recall bias or inflated answers reported by participants. Fourth, the questionnaire was relatively lengthy. It took approximately 10–15 minutes to complete. Finally, similar to the original version, although the ASCRS SPEED II-Thai was designed to recognize OSD prior to lens-based and corneal-based refractive surgeries, it might lack generalizability to asymptomatic cases. Nonetheless, positive questionnaire results could help alert surgeons that patients are likely to have some degree of preoperative OSD. It is strongly recommended to always perform an ocular surface examination, regardless of whether patients screen positive or negative on the questionnaire [5].

In summary, the ASCRS SPEED II-Thai was produced and validated by assessing its validity and reliability. Despite some limitations, the ASCRS SPEED II-Thai is a valid and reliable instrument for ophthalmologists to preoperatively screen for OSD in Thai cataract surgery candidates. Due to its adequate psychometric properties, this questionnaire might be applicable in routine clinical practice, complimenting clinical examination and objective OSD tests with potentially measurable patient benefits.

## Author Contributions

**Conceptualization:** Kaevalin Lekhanont, Paphada Ariyanonthacha, Chathaya Wongrathanandha.

**Data curation:** Kaevalin Lekhanont, Paphada Ariyanonthacha, Chathaya Wongrathanandha, Prae Phimpho, Nontawat Cheewaruangroj.

**Formal analysis:** Kaevalin Lekhanont, Paphada Ariyanonthacha, Chathaya Wongrathanandha.

**Investigation:** Kaevalin Lekhanont, Paphada Ariyanonthacha, Prae Phimpho, Nontawat Cheewaruangroj.

**Methodology:** Kaevalin Lekhanont, Paphada Ariyanonthacha, Chathaya Wongrathanandha, Prae Phimpho, Nontawat Cheewaruangroj.

**Project administration:** Paphada Ariyanonthacha.

**Resources:** Chathaya Wongrathanandha.

**Supervision:** Kaevalin Lekhanont.

**Validation:** Chathaya Wongrathanandha.

**Writing – original draft:** Kaevalin Lekhanont, Paphada Ariyanonthacha.

**Writing – review & editing:** Kaevalin Lekhanont, Chathaya Wongrathanandha, Prae Phimpho, Nontawat Cheewaruangroj.

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
