## [Decision Letter · Decision Letter 0]

11 Mar 2024

Validity and Reliability of the Thai Version of the ASCRS SPEED II© Preoperative OSD Questionnaire in Thai Cataract Surgery Candidates

PONE-D-23-33929

Dear Dr. Lekhanont,

We’re pleased to inform you that your manuscript has been judged scientifically suitable for publication and will be formally accepted for publication once it meets all outstanding technical requirements.

Kind regards,

Manuel Garza León

Academic Editor

PLOS ONE

Journal Requirements:

Additional Editor Comments (optional):

Reviewers' comments:

Reviewer's Responses to Questions

**Comments to the Author**

1. Is the manuscript technically sound, and do the data support the conclusions?

Reviewer #1: Yes

2. Has the statistical analysis been performed appropriately and rigorously? 

Reviewer #1: Yes

3. Have the authors made all data underlying the findings in their manuscript fully available?

Reviewer #1: Yes

4. Is the manuscript presented in an intelligible fashion and written in standard English?

Reviewer #1: Yes

5. Review Comments to the Author

Reviewer #1: The purpose of this study was to develop a Thai version of the American Society of Cataract and Refractive Surgery (ASCRS)-modified Standard Patient Evaluation of Eye Dryness (SPEED) II© preoperative ocular surface disease (OSD) questionnaire (ASCRS SPEED II-Thai) and evaluate its validity and reliability in Thai cataract surgery candidates.

A few comments should be noted:

1. Content validity index (CVI) for each item and the overall questionnaire was performed correctly to assess the content validity of the questionnaire.

2. The statistical analysis is sounds and reflects the key elements to assess Validity and Reliability.

3. The discussion successfully deliver the context and the relevant previous studies, and lands a coherent conclusion taking into account the results obtained.

6. PLOS authors have the option to publish the peer review history of their article (what does this mean?). If published, this will include your full peer review and any attached files.

Reviewer #1: **Yes: **Roberto Gonzalez-Salinas
